# A Phase II Study on the Use of Convalescent Plasma for the Treatment of Severe COVID-19- A Propensity Score-Matched Control Analysis

**DOI:** 10.3390/microorganisms9040806

**Published:** 2021-04-11

**Authors:** Vasiliki Pappa, Anthi Bouchla, Evangelos Terpos, Thomas P. Thomopoulos, Margherita Rosati, Dimitris Stellas, Anastasia Antoniadou, Andreas Mentis, Sotirios G. Papageorgiou, Marianna Politou, Anastasia Kotanidou, Ioannis Kalomenidis, Garyfalia Poulakou, Edison Jahaj, Eleni Korompoki, Sotiria Grigoropoulou, Xintao Hu, Jenifer Bear, Sevasti Karaliota, Robert Burns, Maria Pagoni, Ioannis Trontzas, Elisavet Grouzi, Stavroula Labropoulou, Kostantinos Stamoulis, Aristotelis Bamias, Sotirios Tsiodras, Barbara K. Felber, George N. Pavlakis, Meletios- Athanasios Dimopoulos

**Affiliations:** 1Hematology Unit, Second Propaedeutic Department of Internal Medicine and Research Institute, School of Medicine National and Kapodistrian University of Athens, University General Hospital “Attikon”, 18120 Athens, Greece; anthibouhla@hotmail.com (A.B.); th.thomopoulos@gmail.com (T.P.T.); sotirispapageorgiou@hotmail.com (S.G.P.); abamias@med.uoa.gr (A.B.); 2Department of Clinical Therapeutics, School of Medicine, National and Kapodistrian University of Athens, 11528 Athens, Greece; eterpos@med.uoa.gr (E.T.); e.korompoki@imperial.ac.uk (E.K.); mdimop@med.uoa.gr (M.-A.D.); 3Human Retrovirus Section, Vaccine Branch, Center for Cancer Research, National Cancer Institute, Frederick, MD 21702-1201, USA; margherita.rosati@nih.gov (M.R.); dimitrios.stellas@nih.gov (D.S.); sevasti.karaliota@nih.gov (S.K.); george.pavlakis@nih.gov (G.N.P.); 4National Hellenic Research Foundation, Institute of Chemical Biology, 11635 Athens, Greece; 5Fourth Department of Internal Medicine, University General Hospital “Attikon”, School of Medicine, National and Kapodistrian University of Athens, 11527 Athens, Greece; Ananto@med.uoa.gr (A.A.); grigoropoulou.sotiria@gmail.com (S.G.); tsiodras@med.uoa.gr (S.T.); 6National Influenza Reference Laboratory of Southern Greece, Hellenic Pasteur Institute, 11521 Athens, Greece; mentis@pasteur.gr (A.M.); vlabropoulou@pasteur.gr (S.L.); 7Hematology Laboratory-Blood Bank, Aretaieion Hospital, School of Medicine, National and Kapodistrian University of Athens, 11528 Athens, Greece; mariannapolitou@gmail.com; 8First Department of Critical Care Medicine and Pulmonary Services, Evangelismos General Hospital, National and Kapodistrian University of Athens, 11527 Athens, Greece; akotanid@gmail.com (A.K.); ikalom@med.uoa.gr (I.K.); edison.jahaj@gmail.com (E.J.); 93rd Department of Internal Medicine, Sotiria General Hospital, School of Medicine, National and Kapodistrian University of Athens, 11527 Athens, Greece; gpoulakou@gmail.com (G.P.); john-tron@hotmail.com (I.T.); 10Human Retrovirus Pathogenesis Section, Vaccine Branch, Center for Cancer Research, National Cancer Institute, Frederick, MD 21702, USA; xintao.hu@nih.gov (X.H.); jenifer.bear@nih.gov (J.B.); robert.burns@nih.gov (R.B.); barbara.felber@nih.gov (B.K.F.); 11Basic Science Program, Frederick National Laboratory for Cancer Research, Frederick, MD 21701, USA; 12Haematology-Lymphomas Department and BMT Unit, Evangelismos Hospital, 10676 Athens, Greece; marianpagoni@yahoo.com; 13Transfusion Service and Clinical Hemostasis of Saint Savvas, Oncology Hospital of Athens, 11522 Athens, Greece; egrouzi@otenet.gr; 14Hellenic National Blood Transfusion Center, 13678 Athens, Greece; kostas.stamoulis@gmail.com

**Keywords:** convalescent plasma, COVID-19, efficacy, SARS-CoV-2 antibodies

## Abstract

COVID-19 is a global pandemic associated with increased morbidity and mortality. Convalescent plasma (CP) infusion is a strategy of potential therapeutic benefit. We conducted a multicenter phase II study to evaluate the efficacy and safety of CP in patients with COVID-19, grade 4 or higher. To evaluate the efficacy of CP, a matched propensity score analysis was used comparing the intervention (*n* = 59) to a control group (*n* = 59). Sixty patients received CP within a median time of 7 days from symptom onset. During a median follow-up of 28.5 days, 56/60 patients fully recovered and 1 patient remained in the ICU. The death rate in the CP group was 3.4% vs. 13.6% in the control group. By multivariate analysis, CP recipients demonstrated a significantly reduced risk of death [HR: 0.04 (95% CI: 0.004–0.36), *p*: 0.005], significantly better overall survival by Kaplan–Meir analysis (*p* < 0.001), and increased probability of extubation [OR: 30.3 (95% CI: 2.64–348.9), *p*: 0.006]. Higher levels of antibodies in the CP were independently associated with significantly reduced risk of death. CP infusion was safe with only one grade 3 adverse event (AE), which easily resolved. CP used early may be a safe and effective treatment for patients with severe COVID-19 (trial number NCT04408209).

## 1. Introduction

The SARS-CoV-2 coronavirus outbreak, which first occurred in Wuhan, China, on 12 December 2019, is now a global threat. The SARS-CoV-2 virus causes a severe form of infection called corona-virus disease 2019 (COVID-19) [1,2]. The SARS-CoV-2 virus is a b coronavirus and has an 84% nucleic acid homology to the Chinese Horseshoe bat, 78% similarity with SARS-CoV and 50% with MERS-CoV [3]. The four structural genes of SARS-CoV-2 encode the nucleocapsid protein N, the spike protein S, the small membrane protein SM, the membrane glycoprotein N and an additional membrane glycoprotein HE [4]. 

Similar to other viruses, SARS-CoV-2 infects the pulmonary alveolar epithelial cells by endocytosis using the receptor of the angiotensin II converting enzyme (ACE II) [5].

In 80% of cases the disease is mild, but in some patients, especially in patients with comorbidities, a severe form of the disease develops, with increased mortality associated with complications such as acute respiratory distress syndrome [6,7] and cardiovascular and thromboembolic events [8,9,10,11].

Until now, only three agents have shown some clinical efficacy in large randomized controlled trials, namely remdesivir for hospitalized patients with pulmonary involvement, dexamethasone in hospitalized patients in need of oxygen support and colchicine for moderate to severe disease, reducing the length of oxygen therapy and hospitalization [12,13,14]. In addition, passive immunization of patients using convalescent plasma (CP) from individuals fully recovered from COVID-19 [15] is a therapeutic strategy with potential benefit. The administration of CP or hyperimmune globulins (hyper-IG) from patients recovered from other viral infections, i.e., SARS, MERS, Influenza A H1N1 and Ebola virus, has been used in the past, resulting in reduction of the duration of hospitalization and reduction of mortality [16,17,18,19]. CP infusion transfers antibodies against the above-described viral proteins capable of neutralizing the virus; it also exerts immunomodulatory effects like neutralization of cytokines, complements and autoantibodies and may also activate immune cells like dendritic cells as well as T and B-cells [20].

Published data on the use of CP for the treatment of COVID-19 are gradually increasing, with various results depending on the design of the trials and the population of the patients [21,22,23,24,25,26,27,28,29,30,31,32,33,34,35,36,37,38,39,40,41,42,43,44,45,46,47]. Recently, the FDA modified the Emergency Use Authorization of CP [48] to the use of high titer CP for the treatment of hospitalized patients with COVID-19 early in the disease course and for hospitalized patients with impaired humoral immunity who cannot produce an adequate antibody response [49].

Taking into consideration discrepancies in the literature about the efficacy of CP infusion in severe COVID-19, we conducted a phase II multicenter study aimed at investigating the efficacy of CP for the treatment of hospitalized patients with severe COVID-19, regarding overall survival on day 28 as well as the safety of the treatment and its effect on clinical improvement like duration of hospitalization, of stay in the ICU and of oxygen support. We compared the outcomes to a matched control group of patients treated in the same hospitals during the same time period not receiving the intervention and treated according to the standard of care.

We present here the outcome of CP infusion on the first 60 patients with COVID-19 treated with CP and compare the results of 59 CP recipients to 59 controls using a matched propensity score analysis.

## 2. Materials and Methods

### 2.1. Study Design

This study is a multicenter ongoing prospective phase II trial (identifier number NCT04408209), conducted at 5 hospitals in Athens, Greece. All study procedures were carried out in accordance with the declaration of Helsinki (18th World Medical Association Assembly), its subsequent amendments, Greek regulations and guidelines, as well as the good clinical practice guidelines (GCP) as defined by the International Conference of Harmonization. The study was also approved by the local ethics committees of all participating hospitals. All patients provided written informed consent.

The primary endpoint was survival on day 28. The secondary endpoints were: time to clinical improvement (i.e. patients not fulfilling the criteria for severe disease), safety, duration of hospitalization, duration of stay in the ICU, duration of ventilation support/ECMO if applicable, and time until negative SARS-CoV-2 PCR (nasal/pharyngeal swab). Additional analyses performed included the predictive value of comorbidities and inflammation markers on mortality, the titer of anti-SARS-CoV-2 antibodies in the infused plasma units, and investigation of the titer of anti-SARS-CoV-2 antibodies in the patients before the infusion of CP on days 1–7 and weekly until day 35.

### 2.2. Patients’ Inclusion Criteria and Longitudinal Analyses Performed

From 7 May 2020 to 10 November 2020, 60 patients with ≥ grade 4 COVID-19 disease according to WHO criteria were enrolled in the study and received CP transfusion. The diagnosis was confirmed by real-time RT-PCR assay of the nasopharyngeal swab. Inclusion criteria included: (1) age > 18 years; (2) confirmed COVID-19 by PCR; (3) symptom onset less than 10 days prior; (4) severe disease as shown by one of the following: (i) respiratory rate 30 min; (ii) Hb SAT 93% (FiO2 = 0.21); (iii) CRP > 1.5 (NR < 0.4) or > 3x UNL; ferritin > 100 ng/mL; (iv) PaO2:FiO2 < 300 mg; (v) pulmonary infiltrates on CT scan or chest X-ray; (5) life-threatening disease as determined by one of the following: (i) respiratory failure; (ii) septic shock; (iii) multiorgan failure; (iv) intubation duration < 72 h; (6) signed, informed consent by either the patient or the patient’s legal representative in the case of intubated patients.

Patients fulfilling criteria 1, 2, 3, 6 and one of either 4 or 5 were included. The control group included patients hospitalized during the same time period and with similar disease characteristics at the time of admission but who did not sign informed consent to receive CP; these patients were included only so their data could be analyzed. Clinical and laboratory parameters were registered for the first 7 days for the CP recipients only and on a weekly basis thereafter until day 28. Anti-SARS-CoV-2 antibody titers were determined in the recipients only on days 1–7 and on days 14, 21 and 28.

A real-time one–step reverse transcription–PCR, specific for the ORF1ab gene of SARS-CoV-2 and for the N gene of all other coronaviruses, from the nasopharyngeal swab was performed on days 1, 4, 7, 10, 14, 21, and 28 in the CP recipients using the VIASURE SARS-CoV-2 Real Time PCR Detection Kit (CerTest Biotec SL, Zaragoza, Spain). The Ct values reflecting the number of cycles needed for the first detection of the viral RNA during the real-time PCR reaction were used as an indirect indication of the viral load (higher Ct values reflected lower viral load).

### 2.3. CP Infusion Treatment Protocol

All patients received treatment with single-donor CP, ABO identical, that included the infusion of 200–233 mL of CP in 30–60 min on days 1, 3 and 5. The CP stored as fresh frozen plasma, negative for HBV, HCV, HIV, VDRL, and HTLV-1, was infused within 1 h after thawing.

### 2.4. CP Donors

Individuals who had recovered from SARS-CoV-2 infection were invited to donate plasma after written informed consent was obtained. Criteria for plasma donors’ inclusion were previously described [50].

### 2.5. Detection of Anti-SARS-CoV-2 Antibodies in the Donors and Recipients

We used two methods for the detection of anti-SARS-CoV-2 antibodies in putative plasma donors, as previously described [50]. The main method used for making the decision to proceed to plasmapheresis was a commercially available ELISA (Euroimmun Medizinische Labordiagnostika AG) that detects IgG and IgA antibodies against the recombinant S1 domain of the Spike protein of the virus (S1 domain), as previously described [50]. The results were interpreted as positive if the index value was >1.1 optical density (OD), negative if <0.9 OD, and borderline between 0.8 and 1.1 OD. This method was also used for the detection of anti-SARS-CoV-2 antibodies in the plasma recipients during the course of the disease.

In both donors and recipients, we also performed (i) an in-house ELISA to detect either the complete Spike (amino acid (AA) 15-1208_2P) or Spike-RNA binding domain (Spike_RBD) (AA 319-525) using mammalian Expi293-cell-produced proteins, or E. coli-produced complete Nucleocapsid protein (N) or its RNA binding domain (N-RBD, AA 47-173) and (ii) a neutralizing antibody (NAb) assay using SARS-CoV-2 pseudotyped virus, as previously described [50,51].

### 2.6. Statistical Analysis

A matched propensity score analysis was performed to select the most suitable controls for the intervention group. A 1:1 ratio without replacement was used. The factors selected for matching were age, gender, baseline SOFA score, time from symptom onset to diagnosis, and concomitant dexamethasone use. A standardized difference below 0.3 after the matching process was considered acceptable. After matching, the baseline characteristics of the control group were compared to the intervention group using non-parametric tests, as appropriate. For reasons of comparability, day 1 was defined as the day of hospital admission, for both the intervention and the control group for all parameters analyzed, except for the longitudinal analyses performed in the CP group, where day 1 was defined as the first day of CP infusion.

Univariate Cox proportional hazard regression models were used to evaluate time-dependent outcomes, namely time to death, time to exit from ICU, time to intubation and extubation, time to hospital discharge, and time to achievement of SARS-CoV-2 PCR negativity. Regarding overall survival, variables that were found to be statistically significant in the univariate analysis were included in a multivariate Cox regression model. A subgroup analysis regarding the primary endpoint by the level of antibodies in the infused CP was also performed. Univariate binary logistic regression was used to assess the aforementioned outcome irrespective of time. Kaplan-Meier analysis was used to evaluate cumulative incidence as a function of time. The log-rank Mantel-Cox test was used to test for statistically significant differences of survival. Clinical status on day 14, 28 and at the end of follow-up was evaluated with univariate ordinal logistic regression analysis. The respective variable consisted of four categories, namely death, hospitalized in ICU, hospitalized, and discharged from hospital.

All continuous variables including antibody levels were summarized as median and interquartile range (IQR), assuming deviation from normality. Categorical variables were constructed using the median as cut-off. Antibody levels among different subgroups were compared using Kruskal-Wallis test and Mann-Whitney-Wilcoxon tests. Univariate and multivariate binary logistic regression analyses with the antibody levels as a dependent variable were used to find predictors of antibody response in donors and recipients.

Laboratory variables, including lymphocytes, platelets, C-reactive protein (CRP), ferritin, fibrinogen, LDH, IL-6, SARS-CoV-2 Ct values, SOFA score as previously described [52], and antibody levels on days 2, 7, 14, 21 and 28 were compared to the respective variables on day 1 using non-parametric Wilcoxon test for related samples. For these variables, day 1 was defined as the day of first CP infusion, and all measurements on day 1 were conducted prior to the CP infusion. The trend of clinical and laboratory variables as well as antibody levels over time were evaluated fitting a generalized linear model using generalized estimating equations. Assuming an asymmetrical distribution of variables, logarithmic transformation was performed. The effect of the antibody levels in the infused plasma on the trend of each variable was evaluated incorporating an antibody*time interaction term in the respective model.

All statistical analyses were performed using SPSS version 23. Matched propensity score analysis was performed using SAS.

## 3. Results

### 3.1. CP Donors

60 units of CP were collected by plasmapheresis from 59 patients (36 males and 23 female); one patient had undergone two consecutive plasmapheresis sessions, as she was found to have high antibodies during follow-up. Median age was 46 years (IQR: 22), and median time from symptom onset to plasmapheresis was 61.5 days. All donors were positive for anti-SARS CoV-2 antibodies on the day of plasmapheresis; median level of anti-S1 IgA was 6.13 (IQR: 5.35) and median level of IgG antibodies was 3.42 (IQR: 5.37), using the Euroimmun ELISA. The respective medians for antibodies according to the in-house ELISA were Spike 4.77 (IQR: 2.07), Spike_RBD 3.96 (IQR: 2.40), N_RBD 2.94 (IQR: 2.51), and Neutralizing Abs ID50 2.48 (IQR: 1.49).

### 3.2. CP Recipients: Clinical Characteristics

From 7 May 2020 to 10 November 2020, 60 patients with WHO grade ≥ 4 COVID-19 disease were enrolled and received CP transfusion. A 67-year-old male with multiple myeloma, who received one dose of CP following intubation and succumbed to the infection the following day was excluded from the comparative analyses because a matched control patient could not be found. Patient characteristics at diagnosis are shown in Table 1. Median age was 59 years (IQR: 18 years). Median time from symptom onset to hospital admission and CP transfusion was three days and seven days, respectively. Antibacterial treatment and dexamethasone were used in 59.3% of the patients, whereas remdesivir and hydroxychloroquine were used in 5.1% and 3.4% of the patients, respectively.

Regarding oxygen support, 15% of the patients were on room air, 43.3% were on nasal cannula, 31.7% on venturi mask and 10% on mechanical ventilation.

On computer-assisted tomography (CT), all patients showed bilateral ground-glass opacities and/or pulmonary parenchymal consolidation with predominantly subpleural and bronchovascular bundle distribution. The percentage of infiltrates in the baseline CT scan is shown in Table 1.

### 3.3. Control Group

Records of all patients who were diagnosed during the same time period and hospitalized in the same tertiary hospitals as the CP recipients were retrospectively obtained. Thus, 144 controls were included in the matched propensity score analysis. Matching according to age, gender, baseline SOFA score, time from symptom onset to diagnosis, and concomitant use of dexamethasone resulted in the exclusion of 85 controls. The remaining 59 controls were included in the final analysis. As shown in Table 1, comparison of baseline characteristics and concomitant medication between the intervention and control groups yielded no statistically significant differences.

### 3.4. Outcomes

Regarding primary outcome after a median follow-up of 28.5 days, comparing the 59 CP recipients to the control group, 57/59 recipients (98.3%) remained alive. Fifty-six patients recovered completely and were discharged from hospital after a median length of hospital stay of 15 days, whereas one patient remained intubated in the ICU. Regarding the two deaths in the CP group (3.4%), these included an 82-year-old female with a history of dementia and hypertension who was intubated on day 2 after CP infusion and died of bacterial sepsis after 20 days and a 69-year-old male with a history of hypertension who was intubated on day 2 following CP infusion and died of bacterial sepsis after 66 days.

Regarding the control group, 51/59 (86.4%) patients were discharged after a median hospital stay of 10 days, whereas eight patients (13.6%) died within a median follow-up of 12 days, as shown in Table 2. Sixteen patients of the intervention group were intubated and 13 of them were extubated and discharged from ICU after a median of 15 days. It should be noted that four patients were intubated prior to the CP infusion. Eight controls were intubated; among them, one was extubated and exited the ICU. Comparison of outcomes between the intervention and the control group are summarized in Table 2. Patients in the intervention group had a significantly longer median follow-up time of 29 days vs. 10 days and a longer duration of hospitalization of 15 days vs. 10 days in the control group.

Univariate analysis of factors associated with the primary endpoint demonstrated a statistically significant association between CP and overall survival (OS) (HR: 0.05, 95% CI: 0.01–0.43), as shown in Table 3. The Kaplan-Meier analysis, depicted in Figure 1, also showed a statistically significant association between CP infusion and better OS (Log-rank *p* < 0.001). In a subgroup analysis by the level of antibodies in the infused plasma, no differential effect of antibody levels was found on OS (Table 3). Factors associated with reduced OS were advanced age (HR: 1.08 (95% CI: 1.01–1.14), *p*: 0.024) and the percentage of infiltrates in the CT scan (HR: 2.53 (95% CI: 1.24–5.19), *p*: 0.011), as shown in Table 3. A multivariate model incorporating statistically significant factors, obtained by the univariate analysis, including age and percentage of CT infiltrates and CP infusion, confirmed the independent significant association of CP infusion with better overall survival, as shown in Table 4. Interestingly, on multivariate analysis, infusion of CP with high (above the median) Spike, Spike RBD, N_RBD antibodies or ID50 was associated with improved OS, as opposed to infusion of CP with low antibody levels, where no significant association was noted.

Regarding the association of CP infusion with secondary outcomes, the results of univariate analysis are presented in Table 5. No association was found between CP infusion and clinical status on days 14 and 28 as well as at the end of follow-up. CP infusion was not associated with the risk of intubation or admission to ICU. Finally, CP infusion was not associated with time to reach SARS-CoV-2 PCR negativity. However, a statistically significant association between CP infusion and extubation or exit from ICU was noted (OR: 30.3, 95% CI: 2.64–348.9, OR: 15.16, 95% CI: 2.02–113.3, respectively). High antibody titers in the infused CP predicted a significantly higher rate of extubation and exit from ICU (data not shown). In addition, as shown in Appendix A, advanced age and percentage of infiltrates in the CT scan were associated with worse clinical outcome at the end of follow-up (OR: 1.07 (95% CI: 1.01–1.13), *p*: 0.018 and OR: 2.41 (95% CI: 1.19–4.85), *p*: 0.014, respectively). Factors associated with increased risk for intubation were advanced age (OR: 1.05 (95% CI: 1.01–1.10), *p*: 0.013), percentage of infiltrates in the CT scan (OR: 2.57 (95% CI: 1.47–4.49), *p*: 0.001) and advanced SOFA score [OR: 1.48 (95% CI: 1.19–1.84), *p*: 0.001), as shown in Appendix A.

Subgroup analysis by stratifying recipients according to the time of CP infusion from symptom onset demonstrated no association with any secondary outcome. Sensitivity analysis after the exclusion of patients intubated at enrollment did not change the results. Similarly, sensitivity analyses excluding recipients that received CP after four or seven days did not yield different results. No correlation was found between comorbidities and length of hospital or ICU stay or between the pre-treatment levels of anti-SARS-CoV2 antibodies and disease severity (data not shown).

### 3.5. Adverse Events

One patient had a grade 3 adverse event (AE) consisting of severe exacerbation of dyspnea and hypoxemia after infusion of the first CP dose. The symptoms resolved by conventional measures, and the patient was discharged fully recovered from hospital; however, no subsequent doses of CP on days 3 and 5 were given. All other AEs were grade 1, comprising mild erythema in one patient, mild dizziness in one patient, and increased temperature two hours after first CP infusion in one patient. These AEs were easily handled, and the patients continued the subsequent infusions uneventfully.

### 3.6. Longitudinal Analysis of Clinical and Laboratory Parameters in the CP Group

As shown in Appendix A, median SOFA score declined significantly from five to two on day 7 after CP infusion; however, a slight yet statistically significant decrease was seen even on day 2 from CP infusion. The generalized linear model predicted an average decrease of 25% per week (*p*: 0.02). No interaction was found between the trend of SOFA score change and the level of antibodies in the infused plasma.

The changes of laboratory parameters in the CP recipients, namely lymphocyte and platelet counts as well as CRP, Ferritin, Fibrinogen, LDH and IL-6, are depicted in Appendix A.

Among them, CRP, LDH, and fibrinogen decreased significantly on day 7, whereas a delayed decline in ferritin and IL-6 was observed (on days 14 and 21, respectively). No interaction was found between the level of antibodies in the infused plasma and the trend of inflammatory markers over time; however, high titer of neutralizing antibodies predicted a steeper significant decrease of 17.3% of ferritin (*p* < 0.001) (data not shown).

In addition, regarding viral load, SARS CoV-2 PCR Ct values increased significantly on day 7 (33.1 vs. 26.8, *p* < 0.01) (Appendix A). The generalized linear model predicted an average increase of Ct values by 10% (*p* < 0.001), unaffected by the levels of antibodies in the infused plasma.

Regarding the anti-SARS-CoV-2 antibodies in the recipients, as shown in Figure 2a, a significant increase of anti-S1 IgG and IgA antibodies was observed, starting on day 2 following CP infusion. IgG anti-S1 increased significantly until day 21, and IgA anti-S1 until day 14. As shown in Figure 2b,c, anti-Nucleocapsid, anti-Nucleocapsid_RBD, anti-Spike, anti-Spike_RBD, and Nab peak levels were observed 1–2 weeks post CP, corresponding to 2–3 weeks post symptom onset. The increase of Nab matched the Spike and Spike_RBD antibody increases. No association was found between the level of donor antibodies and antibody trend in the recipients over time by all methods of detection.

### 3.7. Subgroup Analysis by the Level of Antibodies at Baseline

Using the Euroimmun assay at baseline, 31% of patients were positive for anti-S1 IgG antibodies compared to 62.1% positive for anti-S1 IgA (Figure 2a). The probability of detection of positive antibodies at baseline was significantly associated with longer symptom duration, as depicted in Figure 2d. No significant differences of clinical characteristics were observed between patients with and without anti-SARS-CoV-2 antibodies at baseline (data not shown).

Regarding the results of the in-house ELISA at baseline, 26 patients (49%) from the cohort of 53 patients showed no or very low Spike antibody responses (Figure 2c). Seven patients (13%) scored negative for both Spike and Nucleocapsid antibodies.

In subgroup analyses, comparing characteristics and outcomes of patients based on their baseline antibody status by the in-house ELISA assay, positive baseline antibodies were associated with improved clinical outcomes but not with survival. In detail, the presence of Nucleocapsid antibodies at baseline was predictive of improved clinical status on day 7, 14, and 28 (OR: 0.20, 95% CI: 0.05–0.77, OR: 0.21, 95% CI: 0.06–0.71, and OR: 0.20, 95%CI: 0.04–0.92, respectively). Similarly, the presence of Spike antibodies at baseline was predictive of improved clinical status on day 7, 14, 28 (OR: 0.18, 95% CI: 0.05–0.71, OR: 0.13, 95% CI: 0.04–0.47, and OR: 0.18, 95% CI: 0.04–0.76, respectively). Finally, positive Spike-RBD antibodies at baseline were predictive of improved clinical status on days 7 and 14 (OR: 0.25, 95% CI: 0.06–0.95 and OR: 0.24, 95% CI: 0.07–0.83, respectively).

Moreover, the presence of positive Spike antibodies by the in-house ELISA at baseline was predictive of a decreased risk or intubation and admission to ICU (OR: 0.22, 95% CI: 0.06–0.88, and OR: 0.22, 95% CI: 0.06–0.88), respectively.

Importantly, patients with negative baseline anti-S1-IgG experienced a significantly steeper increase of IgG antibodies between days 1–7 by 22% (*p* < 0.001); similarly, patients without baseline anti-S1-IgA demonstrated a steeper increase of IgA, by 19% (*p* < 0.001), as shown in Figure 3a,b. In addition, patients with negative baseline antibodies for Nucleocapsid, Nucleocapsid_RBD, Spike_RBD, Spike, and Nabs ID50 demonstrated a steeper increase of these antibodies following CP infusion (Figure 3c–g).

## 4. Discussion

In this report, we present the results of a multicenter phase II study (NCT04408209) from five participating hospitals in Athens, Greece, on the safety and efficacy of CP in 60 patients with at least grade 4 COVID-19 and compare the primary and secondary outcomes to a control group of patients using a matched propensity score analysis.

Regarding the dose of CP and titer of antiviral antibodies in the CP, we did not use any cut-off value, since at the time of designing this study, no data were available regarding this issue. In most clinical trials, one to two units from one or different donors have been proposed for treatment. In some studies, only CP with arbitrarily defined high titers were used, resulting in significant reduction in the risk of death or disease progression [28,29,43]. In the recently published retrospective study based on a US national registry of 3082 patients, the titer of antibodies in CP correlated with clinical outcome, as shown by a reduction of the risk of death within 30 days following high titer CP infusion, but only for non-intubated patients; this shows the efficacy of this regimen early in the disease course [53]. Recently, guidelines for the selection of high titer CP for COVID-19 according to the level of anti-SARS-CoV-2 antibodies based on different assays were issued by the FDA [49]. Importantly, in our study the median level of IgG anti-S1 antibodies in the CP by the Euroimmun assay was 3.42, which is quite close to the value of 3.5 characterizing the high titer CP determined by the Euroimmun Assay according to the FDA guidelines [49].

Another important issue is the optimal time of CP infusion following symptom onset. Indeed, early reports have shown that the administration of CP in critically ill COVID-19 patients showed no significant reduction of mortality [54]. Most importantly, Joyner et al. demonstrated that the 7- and the 30-day mortality rates were significantly increased in patients receiving CP > 4 days from symptom onset [27]. Generally, the time of CP infusion differs significantly in the design of different trials, from 10–22 days [25,33,34,40,44,45,46]. In the study by Altuntas et al., a higher rate of mechanical ventilation support was observed in patients receiving CP 20 days after diagnosis compared to three interval groups (< 5, 6–10, and 11–15 days, *p* = 0.001) [23]. We failed to find a significant effect of the time to CP infusion, regarding all primary and secondary outcomes, in accordance with a recently published randomized trial where no significant difference was observed in mortality or disease deterioration in early (< 7 days of symptom onset) vs. late CP administration [43].

The data regarding the efficacy of CP in COVID-19 are gradually increasing, including small case series [25,34,36], observational studies [27,32,33,42,55], matched controlled studies [21,23,30,37,38,39,40,41,44,45,46,47,56] and a few randomized controlled trials [22,24,26,28,29,35,43], with no definite conclusions. In the observational study by Salazar et al. of 25 patients with severe and life threatening disease, the infusion of CP resulted in the improvement of disease severity in 76% of patients [33]. A single arm multicenter trial from Italy using hyperimmune plasma with neutralizing antibodies titer ≥ 1:160 also showed a mortality rate 6.5% lower than an expected 15% mortality rate according to national statistics [55].

In our study, we examined the beneficial effect of CP in patients with severe COVID-19 using a matched propensity score analysis. This strategy has already been used in other trials, resulting in contradictory results [21,23,30,37,38,39,40,41,44,45,46,47,56]. In our study, the univariate analysis comparing the CP to the control group showed a significantly reduced risk of death. Moreover, the Kaplan–Meir survival analysis revealed a significant difference in OS in favor of the CP group. Importantly, multivariate analysis confirmed that CP infusion was associated with a significantly reduced risk of death. The beneficial effect of CP infusion on survival demonstrated in our study is in accordance with other studies of similar design showing a survival advantage in CP recipients compared to a control group [37,40,41,44]. Some other studies using comparison to a control group have also shown beneficial results in favor of CP for specific subpopulations, including a survival advantage for non-intubated patients [30], a reduction of disease severity for patients with ARDS [39], and a reduction of mortality for elderly—particularly female—patients admitted to ICU and with comorbidities [38]. On the contrary, other studies did not confirm these beneficial findings when comparing the intervention to a control group [45,46,47]. However, in one study, 86% of the patients were intubated and 70% had already high titers of anti-SARS-CoV-2 antibodies before infusion [45]. In other trials, anti-SARS-CoV-2 antibodies were not determined in the CP, which may have interacted with the negative results [46,47]. In one study, the CP was administered within 21 days after symptom onset [46].

Moreover, our findings have not been confirmed by randomized controlled trials [22,26,28,35]. However, several points need to be addressed. The trial published by Li et al. [28] was prematurely closed after approximately 50% of planned patient enrollment, possibly rendering the study underpowered to detect any significant differences between the CP and control arms. The PLACID trial of 464 patients with moderate COVID-19 did not reveal significant differences in mortality or progression of the disease [22]. However, this trial was not blinded, and the antibodies’ titer in the infused plasma was not determined a priori, resulting in 64/160 infused plasmas with undetectable antibodies, which may have interacted with the results. Additionally, and in contrast to other studies [22,32,35] and to ours, no antibody response was observed in the intervention group. In the PlasmAr trial, no significant differences in mortality or clinical outcomes were observed between CP recipients and controls. However, this study involved patients with severe pneumonia but no life threatening disease [35].

Two other randomized trials on CP infusion are available only in a pre-print form. The CONCOVID trial [26] was prematurely closed after the enrollment of the first 86 patients because, at baseline, 53/66 patients had already detectable anti-SARS-CoV-2 neutralizing antibodies. No significant differences in mortality, duration of hospitalization and clinical improvement were observed between the CP and the control group. Another randomized study of 81 patients from Spain was prematurely stopped due to poor recruitment; it was shown that CP could be superior to standard of care [24].

An important finding in our study was the association of higher levels of anti-SARS-CoV-2 antibodies (above the median values) in the infused CP with significantly reduced risk of death. This association is in accordance with the recent publication by Joyner et al., where the death rate was significantly reduced for non-intubated patients receiving CP with high titers of antibodies [53]. However, in our study, due to the small number of patients, we could not demonstrate a discriminative effect between intubated and non-intubated patients.

Importantly, regarding secondary outcomes in our study, CP recipients compared to the control group demonstrated increased probability of extubation and exit from ICU. In addition, high antibody titers in the CP predicted a significantly higher rate of extubation and exit from ICU. These observations extend the beneficial effect of CP to intubated patients. Our findings are in accordance with other studies showing shorter duration in the ICU [29,39], reduction of the recovery time and duration of the infection for patients in the ICU [32] and reduction of mortality for intubated patients [38,44]; in contrast other trials, did not reveal a beneficial effect of CP for intubated patients [40,41,47].

However, CP infusion in our study was not associated with other secondary outcomes, like the clinical status at the end of follow-up, which was significantly associated with advanced age and the percentage of infiltrates in the CT scan indicative of more severe disease. These observations are in line with the results of randomized controlled trials demonstrating no significant differences in the clinical outcome between patients in the CP group and the controls [22,24,26,35] as well as with the results of other non-randomized trials using comparison of CP recipients to matched controls, which failed to show an improvement in time to clinical recovery [41], hospitalization and ventilation times [40,46], or clinical improvement within 28 days [45]. In contrast, other studies of similar design to our study demonstrated in CP recipients an improvement in the supplemental oxygen requirements by day 14 compared to controls [30], improvement in the need for oxygen supply [37], and improvement in the clinical outcome for patients in ARDS [39]. 

Another important observation in our study was that the subgroup of patients negative for anti-SARS-CoV-2 antibodies at baseline showed a more robust antibody increase post infusion. One possible explanation could be that patients without detectable antibodies at baseline, associated with shorter duration of symptoms as shown by our data, had increased viral load, triggering a stronger endogenous antibody immune response, which was further intensified by CP infusion.

Regarding safety, our results are in agreement with other studies of CP infusion showing that it is a safe procedure. In the large trial from the US regarding the FDA expanded Access Program, CP infused in 20,000 hospitalized patients demonstrated low incidence of serious AEs, including transfusion reaction in < 1%, thromboembolic or thrombotic events in <1% and cardiac events in ~3% [57]. No cases of antibody-dependent enhancement (ADE) were found in our study, in accordance with previous reports [25,28,33,34]. ADE represents a well-recognized effect in many viral illnesses [58,59] and is characterized by the facilitation of viral entry into the cells by antibodies or the enhancement of viral toxicity by antibodies [60].

Our study has several limitations. Although the controls were retrospectively selected by propensity score matching, the conclusions drawn are not as robust as through prospective randomized placebo-controlled trials. In addition, the serial changes in laboratory parameters and the antibody response in the control group were not determined since it was a retrospective comparison.

## 5. Conclusions

In conclusion, in this prospective multicenter phase II study, we show through multivariate analysis that CP infusion compared to a matched control group was associated with a significant reduction of the risk of death and a significantly improved overall survival by Kaplan–Meir analysis. Within a median follow-up of 28.5 days, 57/59 patients remained alive and 56 were discharged from hospital fully recovered, with a median hospital stay of 15 days. The death rate in the CP group was 3.4% vs. 13.6% in the control group. At the end of follow-up, 56/59 (94.9%) in the intervention group were discharged compared to 51/59 (86.4%) in the control group; however, this difference was not statistically significant. In addition, 13/59 (22.0%) of patients in the control group exited ICU vs. 2/59 (3.4%) (*p* = 0.014) in the control group. A significant association between CP infusion and extubation or exit from ICU was also noted. High antibody levels in the CP were also associated with significantly improved OS, as shown by multivariate analysis, and with a higher rate of extubation and exit from ICU. CP infusion was safe and side effects were mild and easily managed. These encouraging data need confirmation by randomized controlled trials.

## Figures and Tables

**Figure 1 microorganisms-09-00806-f001:**
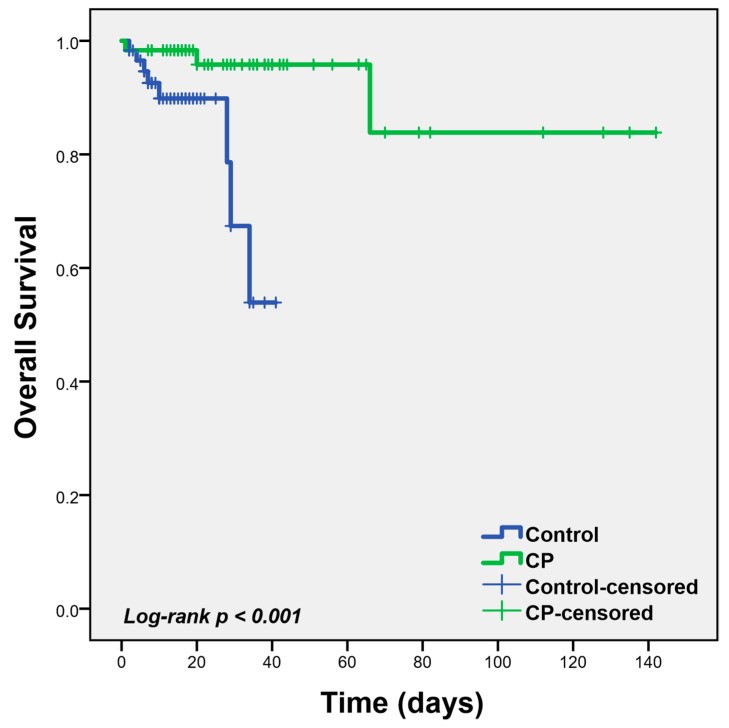
Kaplan-Meier Survival analysis of the recipients and controls.

**Figure 2 microorganisms-09-00806-f002:**
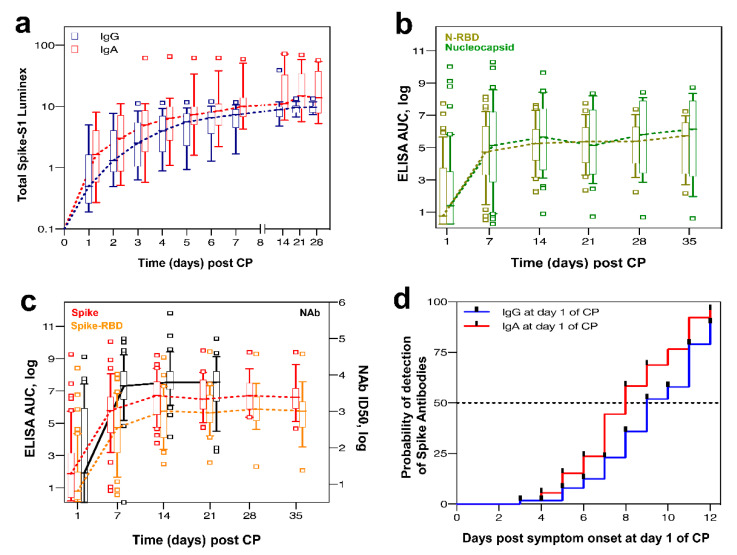
**(****a**) Trend of anti-S1 IgG and IgA antibody levels in the recipients following CP infusion. (**b**) Trends of Nucleocapsid and N_RBD in the recipients following CP Infusion. (**c**) Trend of Spike, Spike_RBD and Nab in the recipients following CP infusion. (**d**) Probability of detection of anti-S1 IgG and IgA in the recipients at day 1 of CP infusion post symptom onset.

**Figure 3 microorganisms-09-00806-f003:**
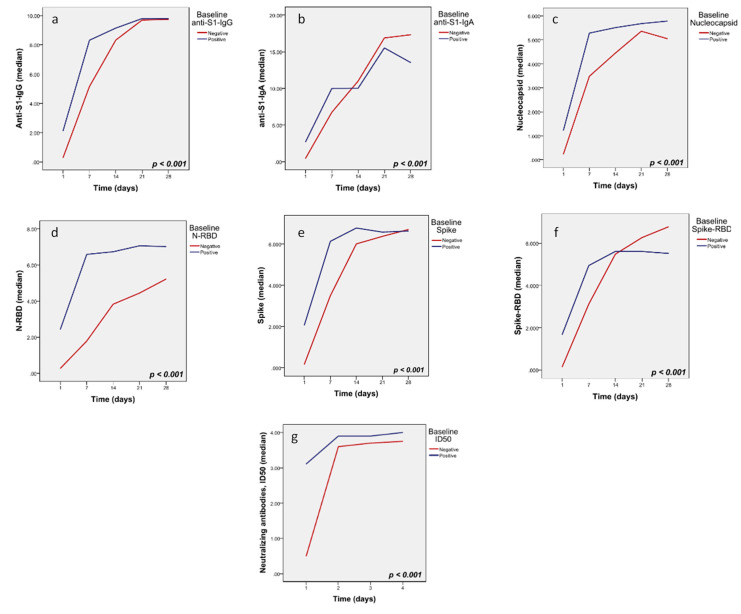
Trend of antibodies by baseline antibody level: (**a**) anti-S1 IgG, (**b**) anti-S1-IgA, (**c**) Nucleocapsid, (**d**) N_RBD, (**e**) Spike, (**f**) Spike_RBD, (**g**) neutralizing antibodies (ID_50_).

**Table 1 microorganisms-09-00806-t001:** Comparison of baseline characteristics of patients in the convalescent plasma and the control group.

	Convalescent Plasma Group (*n* = 59)	Control Group (*n* = 59)	*p*-Value
Age, median (IQR)	59 (18.0)	59 (20)	0.893
<60 years %	45.8	50.8	
≥60 years %	55.2	49.2	
Gender, %			0.564
Female	32.2	37.3	
Male	67.8	62.7	
Comorbidities, %	61.0	62.7	0.393
Diabetes	19.3	28.3	0.269
Arterial hypertension	29.8	37.7	0.382
Coronary artery disease	7.0	13.2	0.282
Heart failure	3.5	9.4	0.205
Pulmonary disease	14.0	9.4	0.457
Renal impairment	8.8	11.3	0.276
Solid tumor	1.8	7.5	0.658
Hematological malignancy	3.5	1.9	0.603
Symptoms, %			
Fever	98.2	96.6	0.571
Myalgia	8.8	10.3	0.775
Cough	56.1	48.3	0.401
Dyspnea	36.8	53.4	0.08
Loss of taste	8.8	3.4	0.235
Anosmia	5.3	5.2	0.983
Diarrhea	19.3	19.0	0.964
Baseline laboratory parameters, median (IQR)			
Lymphocytes, (109/L, NR: 1.1–4.0)	1.17 (0.7)	1.02 (0.6)	0.207
Platelets, (109/L, NR: 130–400)	196 (101.5)	197 (75.3)	0.721
CRP, (mg/L, NR: 0.00–6.00)	47 (50.3)	44.8 (71.9)	0.772
Fibrinogen, (mg/dL, NR: 200–400)	485 (173)	477 (253.9)	0.631
LDH, (U/L, NR: 135–225)	315 (167.8)	277 (127.3)	0.165
Ferritin, (ng/mL, NR: 13–150)	597 (451.5)	474 (167.9)	0.443
Intereukin-6, (pg/mL, NR: <7)	30.5 (43.6)		
SARS-CoV-2 PCR CT value	26.8 (6.9)	27.5 (9.3)	0.700
Percentage of infiltrates at baseline CT, %			0.117
<25	29.1	39.6	
25–50	38.2	43.8	
50–75	25.5	8.3	
≥75%	7.3	8.3	
Concomitant dexamethasone, %	59.3	49.2	0.270
Baseline SOFA score	5 (4)	4 (4)	0.295
Time from first symptom to diagnosis, median (IQR)	3 (4]	4 (3)	0.265
Time from first symptom to CP infusion, median (IQR)	7 (4)		
Time from diagnosis to CP infusion, median (IQR)	3 (3)		

**Table 2 microorganisms-09-00806-t002:** Comparison of outcomes of patients in the convalescent plasma and the control group.

	Convalescent Plasma Group (*n* = 59)	Control Group (*n* = 59)	*p*-Value ^1^
Status at day 14	*n* (%)	*n* (%)	0.249
Discharged	21 (35.6)	31 (52.5)	
Hospitalized	30 (50.8)	18 (30.5)	
In ICU	8 (13.6)	5 (8.5)	
Death	0 (0.0)	5 (8.5)	
Status at day 28	*n* (%)	*n* (%)	0.566
Discharged	48 (81.4)	46 (78.0)	
Hospitalized	5 (8.5)	5 (8.5)	
In ICU	5 (8.5)	3 (5.1)	
Death	1 (1.7)	5 (8.5)	
Status at end of follow-up	*n* (%)	*n* (%)	0.106
Discharged	56 (94.9	51 (86.4)	
Hospitalized	0 (0.0)	0 (0.0)	
In ICU	1 (1.7)	0 (0.0)	
Death	2 (3.4)	8 (13.6)	
Follow-up, median (IQR)	29 (24)	10 (11)	**<0.001**
Duration of hospital stay, median (IQR)	15 (10)	10 (11)	**0.006**
Admission to ICU, *n* (%)	16 (27.1)	9 (15.3)	0.116
Exit from ICU, *n* (%)	13 (22.0)	2 (3.4)	**0.014**
Time to exit from ICU, median (IQR)	12.5 (37.25)	7 (NC)	0.824
Intubation, *n* (%)	16 (27.1)	8 (13.6)	0.068
Extubation, *n* (%)	13 (22.0)	1 (1.7)	**0.006**
Time to extubation, median (IQR)	15 (35.5)	17.5 (NC)	0.837
Duration of oxygen support, median (IQR)	7 (11.5)	NA	
Achievement of negative PCR, *n* (%)	37 (62.7)	19 (52.8)	0.167
Time to PCR negativity, median (IQR)	14 (14)	9.5 (14.8)	**0.007**

^1^ Highlighted (bold) *p*-values denote statistically significant results.

**Table 3 microorganisms-09-00806-t003:** Results of univariate Cox regression analysis for the association between the convalescent plasma infusion and the antibody levels in the infused plasma and overall survival.

Variables ^2^	HR (95%CI)	*p*-Value ^1^
Age	1.08 (1.01–1.14)	**0.024**
Male gender	1.22 (0.34–4.35)	0.761
Percentage of infiltrates at CT	2.53 (1.24–5.19)	**0.011**
Baseline SOFA score	1.29 (0.98–1.72)	0.073
Dexamethasone co-medication	1.43 (0.39–5.19)	0.586
Convalescent plasma infusion	0.05 (0.01–0.43)	**0.006**
N_RBD (In-house ELISA), below median	0.07 (0.007–0.76)	**0.029**
N_RBD (In-house ELISA), above median	0.04 (0.002–0.62)	**0.021**
Spike (In-house ELISA), below median	0.07 (0.007–0.77)	**0.029**
Spike (In-house ELISA), above median	0.04 (0.002–0.61)	**0.012**
Spike_RBD (In-house ELISA), below median	0.07 (0.006–0.74)	**0.027**
Spike_RBD (In-house ELISA), above median	0.04 (0.002–0.58)	**0.019**
ID50, below median	0.08 (0.007–0.88)	**0.039**
ID50, above median	0.04 (0.003–0.55)	**0.016**

^1^ Highlighted (bold) *p*-values denote statistically significant results; ^2^ Reference category: no plasma infusion.

**Table 4 microorganisms-09-00806-t004:** Results of multivariate Cox regression analysis for the association between the convalescent plasma infusion and the antibody levels in the infused plasma and overall survival.

Variables	HR (95%CI)	*p*-Value ^1^
Age	1.04 (0.97–1.12)	0.233
Percentage of infiltrates at CT	3.87 (1.56–9.58)	**0.003**
Convalescent plasma infusion	0.04 (0.004–0.36)	**0.005**
Subgroup analyses by level of plasma antibodies (cut -off: median) ^2^		
N_RBD (In-house ELISA), below median	0.08 (0.006–1.09)	0.059
N_RBD (In-house ELISA), above median	0.02 (0.001–0.34)	**0.007**
Spike (In-house ELISA), below median	0.10 (0.008–1.21)	0.070
Spike (In-house ELISA), above median	0.02 (0.001–0.33)	**0.007**
Spike_RBD (In-house ELISA), below median	0.08 (0.007–1.003)	0.051
Spike_RBD (In-house ELISA), above median	0.02 (0.001–0.35)	**0.008**
ID50, below median	0.14 (0.01–1.91)	0.139
ID50, above median	0.02 (0.001–0.29)	**0.016**

^1^ Highlighted (bold) *p*-values denote statistically significant results; ^2^ Reference category: no plasma infusion.

**Table 5 microorganisms-09-00806-t005:** Results of univariate regression analyses for the association between the convalescent plasma infusion and secondary outcomes.

Variables	Effect Estimate (95%CI)	*p*-Value ^#^
Clinical status at day 14	OR: 1.50 (0.76–2.98)	0.244 ^1^
Clinical status at day 28	OR: 0.77 (0.31–1.88)	0.565 ^1^
Clinical status at end of follow-up	OR: 0.33 (0.08–1.33)	0.119 ^1^
Hospital discharge	OR: 2.93 (0.74–11.64)	0.127 ^2^
Time to hospital discharge	HR: 0.68 (0.46–0.99)	**0.05 ^3^**
Intubation	OR: 2.37 (0.93–6.01)	0.072 ^2^
Time to intubation	HR: 0.48 (0.19–1.21)	0.122 ^3^
Extubation	OR: 30.3 (2.64–348.9)	**0.006 ^2^**
Time to extubation	HR: 0.68 (0.08–5.44)	0.712 ^3^
Exit from ICU	OR: 15.16 (2.02–113.3)	**0.008 ^2^**
Time to exit from ICU	HR: 0.54 (0.07–4.41)	0.566 ^3^
Achievement of PCR negativity	OR: 1.84 (0.78–4.36)	0.168 ^2^
Time to reach PCR negativity	HR: 0.74 (0.42–1.29)	0.741 ^3^

^#^ Highlighted (bold) *p*-values denote statistically significant results ^1^ Ordinal logistic regression analysis; ^2^ binary logistic regression analysis; ^3^ cox proportional hazard regression analysis.

## Data Availability

The data presented in this study are available on request from the corresponding author.

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
