# Peer review of "A Phase II Study on the Use of Convalescent Plasma for the Treatment of Severe COVID-19- A Propensity Score-Matched Control Analysis"

_microorganisms, 2021, doi:10.3390/microorganisms9040806_

Round 1

Reviewer 1 Report

This is indeed an interesting study of Pappa and coworkers and the topic is hot since it is directly linked with the COVID-19 pendamic. The authors of the work have have shown by multivariate analysis that CP infusion compared to a matched control group can be associated with a significant reduction of the risk of death and a significantly improved overall survival by Kaplan Meir analysis. Within a median follow up of 28.5 days, it was observed that 57/59 patients 567 remain alive and 56 were discharged of hospital fully recovered with a median hospital stay of 15 days. However, the graphics provided are not impressive and of low resolution quality. The objective of the paper is not very transparent, which should be modified. The discussion section is also too long, which should be shorten to a maximum of two pages. Background references are not up to mark. 

Reviewer 2 Report

Today very important is finding of good and effective drug against COVID-19. Many researchers test convalescent plasma. Authors in this paper presented that convalescent plasma significantly reduces risk of death during COVID-19, and when used early may be a safe and effective in patients with severe disease. I suggest two corrections:

  1. In Introduction, should be more about viral proteins and antibodies produced by human body. You can cite article https://pubmed.ncbi.nlm.nih.gov/33408775/ in which are described external SARS-CoV-2 proteins. This will help the reader understand what convalescent plasma should lead during treatment.
  2. In Discussion you can add some works with lack of activity of convalescent plasma.
